# SulfoxFluor-enabled deoxyazidation of alcohols with NaN$_3$

Junkai Guo[1,2], Xiu Wang[1,2], Chuanfa Ni [1], Xiaolong Wan[1] & Jinbo Hu [1✉]

Direct deoxyazidation of alcohols with NaN$_3$ is a straightforward method for the synthesis of widely used alkyl azides in organic chemistry. However, known methods have some limitations such as high reaction temperatures and narrow substrate scope. Herein, a general and practical method for the preparation of alkyl azides from alcohols using NaN$_3$ has been developed. *N*-tosyl-4-chlorobenzenesulfonimidoyl fluoride (SulfoxFluor) plays an important role in this deoxyazidation process, which converts a broad range of alcohols into alkyl azides at room temperature. The power of this deoxyazidation protocol has been demonstrated by successful late-stage deoxyazidation of natural products and pharmaceutically relevant molecules.

[1] Key Laboratory of Organofluorine Chemistry, Center for Excellence in Molecular Synthesis, Shanghai Institute of Organic Chemistry, University of Chinese Academy of Sciences, Chinese Academy of Sciences, 345 Ling-Ling Road, Shanghai 200032, China. [2] These authors contributed equally: Junkai Guo, Xiu Wang. ✉email: jinbohu@sioc.ac.cn

Organic azides are a class of important compounds that have been used as precursors for nitrenes and in the synthesis of amines, and more popularly in the copper-catalyzed azide-alkyne cycloadditions (known as click chemistry)[1–10]. Alkyl azides are typically prepared by nucleophilic substitution ($S_N2$) with an azide ion ($N_3^-$), and the diazo-transfer reaction to primary amines using triflyl azide ($CF_3SO_2N_3$) or fluorosulfuryl azide ($FSO_2N_3$) has emerged as a powerful method for the preparation of alkyl azides from primary amines[1,11]. On the other hand, given the ready availability of structurally diverse alcohols, direct conversion of alcohols to alkyl azides via deoxyazidation would be an attractive synthetic strategy, which avoids the use of genotoxic alkyl halides and sulfonates in azidation reactions[12]. Unfortunately, the alcoholic hydroxyl group is a poor leaving group, so its direct displacement by azide ion is generally difficult. Previously, Mitsunobu conditions have been investigated by using different azide ion sources such as $HN_3$, $TMSN_3$, $(PhO)_2P(O)N_3$, $Zn(N_3)_2\cdot2Py$, or $n\text{-Bu}_4NN_3$, but the Mitsunobu conditions are not amenable to the most readily available and cost-effective azide source—$NaN_3$ (Fig. 1, Eq 1)[13–26]. Indeed, the currently known $NaN_3$-based deoxyazidation methods are sparse (Fig. 1, Eq 2). Both $NaN_3/BF_3\cdot Et_2O$[27] and $NaN_3/$triphosgene[28] systems are only applicable to allylic and benzylic alcohols, and other $NaN_3$-based methods (using $NaN_3/TsIm$[29], $(2,4\text{-}Cl_2C_6H_3O)_2P(O)Cl/NaN_3$[30], or halocarbon/$Ph_3P/NaN_3$[31–34]) suffer from the high reaction temperatures and/or narrow substrate scope. Therefore, the development of a general method for efficient and direct conversion of alcohols to alkyl azides with $NaN_3$ is highly desirable.

Sulfonimidoyl compounds possess diverse reactivity (compared with sulfonyl compounds) due to the modulation by the nitrogen substituent[35–37]. During the past decade, our group has been interested in developing fluoroalkyl sulfoximines as versatile fluoroalkylation reagents[38,39]. Recently, we reported that N-tosyl-4-chlorobenzenesulfonimidoyl fluoride (SulfoxFluor) can serve as a deoxyfluorination reagent for converting alcohols to alkyl fluorides (Fig. 1, Eq 3)[40]. In this fluorination process, the in situ formed alkyl sulfonimidate (from SulfoxFluor and alcohol) serves as the real electrophilic alkylating agent to react with hydrogen-bonded fluoride ion, affording the desired alkyl fluoride[40,41]. We envisioned that since fluoride ion is a weak nucleophile[42], if there is a strong nucleophile (namely, azide ion) existing in the reaction system, the deoxyazidation of alcohol could become the dominating reaction pathway, giving an alkyl azide as a major product.

Herein, we show a general and practical protocol for deoxyazidation of readily available alcohols with $NaN_3$ using SulfoxFluor as an activator (Fig. 1, Eq 4). A wide range of alkyl azides could be obtained successfully under mild reaction conditions.

## Results

**Optimization of reaction conditions.** At the onset of our investigation, we chose the primary alcohol **2a** as a model substrate, $NaN_3$ as an azidation agent, SulfoxFluor as an activator, 1,8-diazabicyclo-[5.4.0]undec-7-ene (DBU) as a base[40,43–46], and DMF as a solvent; and the reaction was carried out at room temperature. The preliminary result showed that the use of SulfoxFluor (1.0 equiv) afforded azide **3a** in 59% yield and a majority of **2a** (31%) remained (Table 1, entry 1). Further optimization of the equiv of SulfoxFluor, $NaN_3$, and DBU showed that azide **3a** was formed in 84% yield without fluorination and elimination by-products (Table 1, entry 4). Reducing the equiv of $NaN_3$ resulted in the formation of alkyl fluoride **4** (Table 1, entries 5–6). No azide **3a** was formed when triethylamine and pyridine were used as bases (Table 1, entries 7–8). For secondary alcohol **2b**, it was found that the use of 2.2 equiv of SulfoxFluor was not enough, and the desired alkyl azide **3b** was formed in only 38% yield, along with a significant amount of **2b** (44%) remained (Table 2, entry 1). Changing the amounts of both SulfoxFluor (2.8 equiv) and DBU (4.0 equiv) resulted in a higher yield (65%) of **3b** (Table 2, entries 2–4). Further screening of the reaction conditions showed that an 84% yield of **3b** could be obtained in 12 h by performing the reaction with **2b** (1.0 equiv), $NaN_3$ (2.0 equiv), SulfoxFluor (2.8 equiv), and DBU (4.0 equiv) at room temperature; and remarkably, alcohol **2b** was completely consumed and no fluorination and elimination side products were formed (Table 2, entry 5). Notably, the use of perfluorobutanesulfonyl fluoride (instead of SulfoxFluor) resulted in a decrease of the yield of **3b** (68%), with 7% of elimination side product **7** being formed (Table 2, entry 6)[45–47]. Shortening the reaction time to 6 h or using other solvents (such as DMSO, toluene, and $CH_3CN$) did not give better yields of product **3b** (Table 2, entries 7–10).

**Comparison of various sulfonyl fluorides and sulfonimidoyl fluorides in deoxyazidation of alcohols.** To demonstrate the uniqueness of our reagent in the deoxyazidation reaction, several sulfonyl fluorides and sulfonimidoyl fluorides were compared to show their reactivity. 2,2,2-Trifluoroethanol (**2c**) was chosen as a model substrate to react with these reagents under standard conditions. The results are shown in Table 3. An excellent yield of azide **3c** (93%) was formed by using SulfoxFluor as an activator, along with a small amount of **2c** (4%) remained (Table 3, entry 1). Changing the S-substituent to an electron-neutral or more electron-deficient 4-nitrophenyl group resulted in a decrease in the yield of **3c** (Table 3, entries 2 and 3). Moreover, in the case of **1d** with an N-alkyl substituent, no azide **3c** was formed and nearly half of **2c** was converted to the sulfonimidoyl ester intermediate **8d** (Table 3, entry 4). When 2-pyridylsulfonyl fluoride (PyFluor) **1e** and tosyl fluoride **1f** were used, a full conversion to the corresponding sulfonyl ester intermediates was observed (Table 3, entries 5 and 6). Replacing the N-substituent from tosyl to tertiary butyl led to no azide formation, and a recovery of **2c** (82%) was observed (Table 3, entry 7). In the case of perfluorobutanesulfonyl fluoride (PBSF), a lower yield (82%) of **3c** was obtained (Table 3, entry 8); however, PBSF was found to give an elimination by-product as mentioned before (Table 2, entry 6). Finally, when $SO_2F_2$ was used under similar conditions, a low yield (12%) of azide **3c** was formed (Table 3, entry 9). Clearly,

(a) Previous work:

$$R\text{-}OH \xrightarrow{\text{Mitsunobu conditions}} R\text{-}N_3 \qquad (1)$$

[azide ion sources: $HN_3$, $TMSN_3$, $(PhO)_2P(O)N_3$, $Zn(N_3)_2\cdot2Py$, $n\text{-}Bu_4NN_3$, but not $NaN_3$]

$$R\text{-}OH \xrightarrow{\text{NaN}_3\text{-based conditions}} R\text{-}N_3 \qquad (2)$$

[narrow scope and/or high temperature]

$$R\text{-}OH \xrightarrow[\text{DBU, rt, 10-30 min}]{\text{SulfoxFluor}} R\text{-}F \qquad (3)$$

SulfoxFluor

(b) **This work**:

$$R\text{-}OH \xrightarrow{\text{SulfoxFluor, NaN}_3} R\text{-}N_3 \qquad (4)$$

[broad scope and mild conditions]

**Fig. 1 Deoxyazidation of alcohols. a** Illustration of previous work on deoxyazidation of alcohols (Eqs 1–2) and deoxyfluorination of alcohols with SulfoxFluor (Eq 3). **b** Illustration of this work. Eq 4 refers to the SulfoxFluor-mediated deoxyazidation of alcohols with $NaN_3$. Eqs 1–3 refer to the previously reported deoxyazidation of alcohols (previous work), and eq 4 shows the SulfoxFluor-mediated deoxyazidation of alcohols with $NaN_3$ (this work).

**Table 1 Screening of reaction conditions for primary alcohol 2a.**

| Entry[a] | 2a/SulfoxFluor/NaN$_3$/Base | Base | 2a (%)[b] | 3a (%)[b] | 4 (%)[b] | 5 (%)[b] |
|---|---|---|---|---|---|---|
| 1 | 1.0: 1.0: 1.0: 1.0 | DBU | 31 | 59 | 2 | 0 |
| 2 | 1.0: 1.3: 4.0: 1.8 | DBU | 17 | 66 | trace | 0 |
| 3 | 1.0: 1.8: 4.0: 1.8 | DBU | 8 | 79 | 0 | 0 |
| 4 | 1.0: 2.2: 4.0: 1.8 | DBU | trace | 84 | 0 | 0 |
| 5 | 1.0: 2.2: 3.0: 1.8 | DBU | trace | 75 | 8 | 0 |
| 6 | 1.0: 2.2: 2.0: 1.8 | DBU | trace | 69 | 10 | 0 |
| 7 | 1.0: 2.2: 4.0: 1.8 | NEt$_3$ | 92 | 0 | 0 | 0 |
| 8 | 1.0: 2.2: 4.0: 1.8 | pyridine | 90 | 0 | 0 | 0 |

[a]Reactions were conducted on a 0.1 mmol scale.
[b]Yields were determined by $^{19}$F NMR using 1-fluoronaphthalene as an internal standard.

**Table 2 Screening of reaction conditions for secondary alcohol 2b.**

| Entry[a] | 2b/SulfoxFluor/NaN$_3$/DBU | Solvent | 2b (%)[b] | 3b (%)[b] | 6 (%)[b] | 7 (%)[b] |
|---|---|---|---|---|---|---|
| 1 | 1.0: 2.2: 4.0: 1.8 | DMF | 44 | 38 | 0 | 0 |
| 2 | 1.0: 1.3: 4.0: 1.8 | DMF | 59 | 27 | 0 | 0 |
| 3 | 1.0: 2.2: 4.0: 4.0 | DMF | 25 | 50 | 0 | 0 |
| 4 | 1.0: 2.8: 4.0: 4.0 | DMF | 17 | 65 | 0 | 0 |
| 5 | 1.0: 2.8: 2.0: 4.0 | DMF | 0 | 84 | 0 | 0 |
| 6[c] | 1.0: 2.8: 2.0: 4.0 | DMF | 0 | 68 | 0 | 7 |
| 7[d] | 1.0: 2.8: 2.0: 4.0 | DMF | 0 | 82 | 0 | 0 |
| 8 | 1.0: 2.8: 2.0: 4.0 | DMSO | 0 | 81 | 0 | 0 |
| 9 | 1.0: 2.8: 2.0: 4.0 | toluene | 2 | 8 | 42 | 0 |
| 10 | 1.0: 2.8: 2.0: 4.0 | CH$_3$CN | 2 | 17 | 21 | 0 |

[a]Reactions were conducted on a 0.1-mmol scale.
[b]Yields were determined by $^{19}$F NMR using 1-fluoronaphthalene as an internal standard.
[c]Perfluorobutanesulfonyl fluoride (PBSF) was used instead of SulfoxFluor.
[d]The reaction time was 6 h.

SulfoxFluor was superior to other sulfonimidoyl fluorides and sulfonyl fluorides in the present deoxyazidation reaction. It is interesting to note that the use of bis(2,4-dichlorophenyl) chlorophosphate ((2,4-Cl$_2$C$_6$H$_3$O)$_2$P(O)Cl, **1j**)/NaN$_3$/DMAP, a state-of-the-art method for deoxyazidation of alcohols at room temperature[30], failed to convert 2,2,2-trifluoroethanol (**2c**) into azide **3c** (Table 3, entry 10; for details, see the Supplementary Methods).

**Deoxyazidation of alcohols**. With the optimized conditions (Table 1, entry 4 for primary alcohols; Table 2, entry 5 for secondary alcohols) in hand, we investigated the substrate scope of this SulfoxFluor-mediated deoxyazidation reaction using NaN$_3$ as an azide source. The results are shown in Fig. 2. Fifty structurally diverse primary and secondary alcohols were applied in this reaction, nearly half of which are pharmaceutically important molecules. In most cases, the corresponding alkyl azides were obtained in good or excellent yields. The reaction tolerates a variety of functional groups, such as aldehydes, alkenes, alkynes,

ketones, esters, amides, halides, nitro, and sulfonyl groups (see Fig. 2). It has been known that aldehydes are not amenable to Mitsunobu reactions owing to the condensation of the aldehyde functionality with Huisgen zwitterions[48]; however, it is remarkable that under our current azidation reaction conditions, desired product **3w** was obtained in 82% yield. Our reaction is also compatible with the majority of heterocycles; heteroaromatic substrates such as indole, benzothiazole, pyridine, thiazole, thiophene, benzothiophene, and pyrimidine are all suitable substrates for this reaction (see **3k–3o**, **3x**, **3y**, and **3ae**). The reaction of enantiomerically enriched secondary alcohols **2q** and **2v** proceeded smoothly to give products **3q** and **3v** in excellent yields (95% and 96%) and stereospecificity (98.6% and >99.9% e.s.) respectively, which is in accordance with an inversion of configuration resulting from an S$_N$2 mechanism (CCDC 2005774 (**3v**) contains the supplementary crystallographic data for this paper. These data are provided free of charge by The Cambridge Crystallographic Data Centre.)[40,42]. Similarly, the stereogenic centers of **3ab**, **3af**, **3ag**, **3av**, and **3ba** were assigned by analogy. Cyclic

**Table 3 Azidation of CF₃CH₂OH with sulfonyl fluorides and sulfonimidoyl fluorides.**

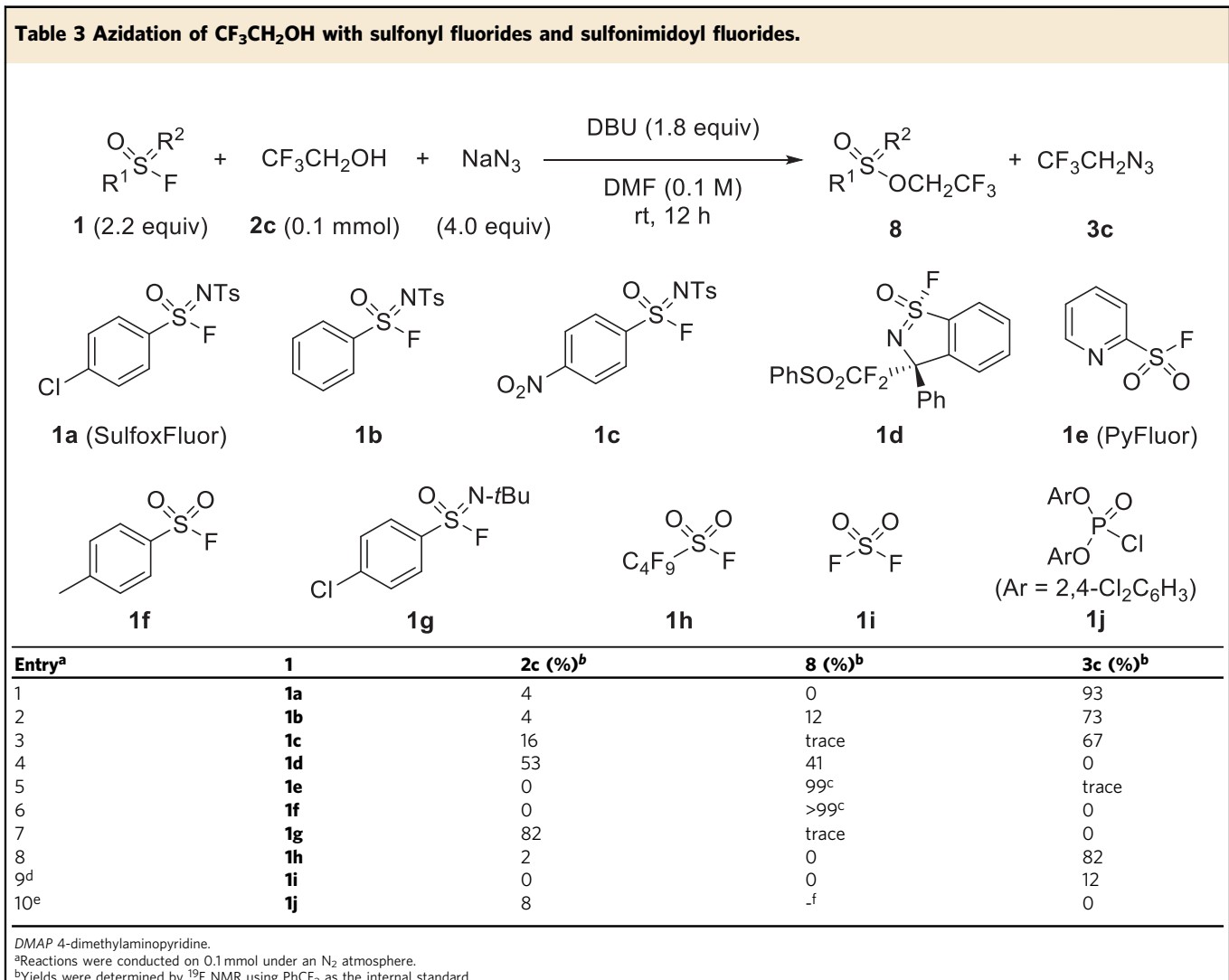

| Entry[a] | 1 | 2c (%)[b] | 8 (%)[b] | 3c (%)[b] |
|---|---|---|---|---|
| 1 | **1a** | 4 | 0 | 93 |
| 2 | **1b** | 4 | 12 | 73 |
| 3 | **1c** | 16 | trace | 67 |
| 4 | **1d** | 53 | 41 | 0 |
| 5 | **1e** | 0 | 99[c] | trace |
| 6 | **1f** | 0 | >99[c] | 0 |
| 7 | **1g** | 82 | trace | 0 |
| 8 | **1h** | 2 | 0 | 82 |
| 9[d] | **1i** | 0 | 0 | 12 |
| 10[e] | **1j** | 8 | -[f] | 0 |

DMAP 4-dimethylaminopyridine.
[a]Reactions were conducted on 0.1 mmol under an N₂ atmosphere.
[b]Yields were determined by ¹⁹F NMR using PhCF₃ as the internal standard.
[c]The existence of **8e** and **8f** was proved by GC-MS.
[d]SO₂F₂ was dissolved in DMF at a concentration of 0.0616 M.
[e]Conditions: **2c** (0.2 mmol), **1j** (1.05 equiv), NaN₃ (4.0 equiv), DMAP (1.2 equiv), DMF (0.2 M), rt, 12 h.
[f]Unidentified products.

alcohols such as four-, five-, and six-membered rings could also undergo efficient deoxyazidation under the standard conditions, giving corresponding products in good to excellent yields (**3u**–**3ae**). Notably, the carbamate group in primary alcohol **2g** is also compatible without elimination and intramolecular cyclization by-products under the present reaction conditions, giving azide **3g** in 60% yield.

Late-stage modification of structurally complex molecules (such as natural products and drugs) can rapidly generate new pharmaceutical candidates with potentially improved pharmacological profiles[49–58]. Late-stage azidation is particularly attractive in this regard because the incorporation of azide functionality (followed by click reaction) can quickly build modular molecular libraries. Inspired by these considerations, we applied our deoxyazidation method in the late-stage functionalization of complex molecules such as isosorbide-2-acetate (**2af**), TF-HF (**2ag**) and (-)-Corey lactone benzoate (**2ah**), and the corresponding azidation products **3af**, **3ag**, and **3ah** were obtained in 92%, 92%, and 81% yields, respectively (Fig. 2). The dansyl chloride derivative **2ai** was also efficiently deoxyazidated to give product **3ai** in 94% yield, indicating that tertiary amine functionality is compatible with the current azidation conditions. It is interesting to mention that, the current deoxyazidation reaction has good selectivity toward multiple alcohols (such as **2ak**-**2an**), that is, the azidation occurs predominantly on the less hindered hydroxyl group and affords the mono-azidation products in good yields (see **3ak**-**3an**). The drug derivatives estrone (**2aq**), oxaprozin (**2ar**), adenosine (**2as**), indometacin (**2at**), bendazac (**2au**), DL-α-tocopherol (**2aw**), estradiol (**2ax**), and glucose derivative **2av** were all able to undergo deoxyazidation smoothly to afford the corresponding products **3aq**-**3ax** in moderate to good yields. Most remarkably, when antifungal drug posaconazole (**2ba**), the most complex molecule among the fifty substrates shown in Fig. 2, was subjected to the current deoxyazidation process, product **3ba** was isolated in satisfactory yield (47%).

Of note that our synthetic method is also advantageous over the previously reported deoxyazidation system (2,4-Cl₂C₆H₃O)₂P(O) Cl/NaN₃/DMAP in converting normal secondary alcohols to organoazides. For example, starting from alcohol **2r**, the use of SulfoxFluor could afford the corresponding azide **3r** in nearly quantitative yield, whereas the use of (2,4-Cl₂C₆H₃O)₂P(O)Cl under reported standard conditions[30] provided **3r** in low yield (21%) even prolonging the reaction time to 12 h. In the latter case, both the bis(2,4-dichlorophenyl) phosphate intermediate and unreacted alcohol **2r** were isolated (for detail, see the Supplementary Methods), indicating the low efficiency of

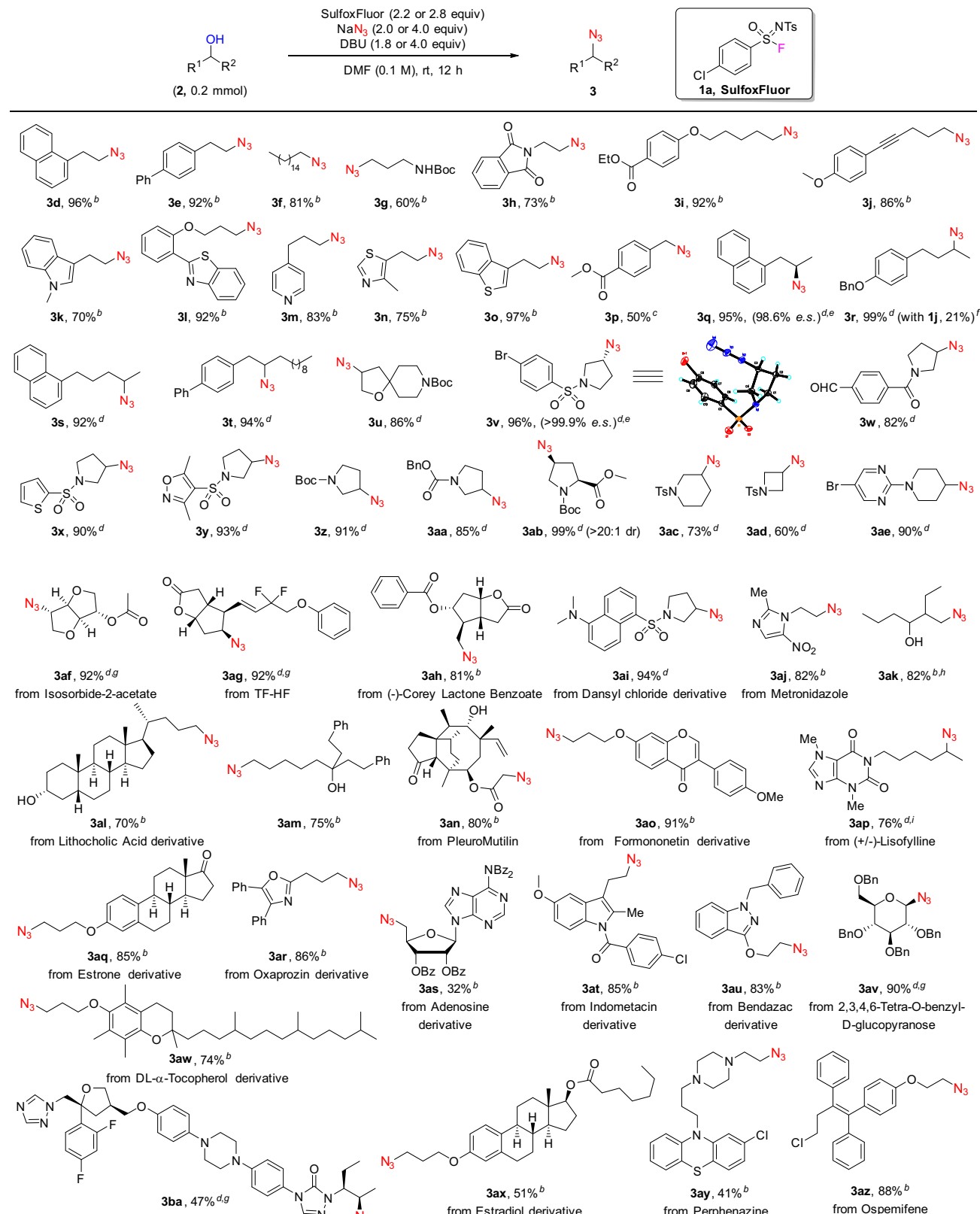

**Fig. 2 Azidation of alcohols using SulfoxFluor[a].** [a]Isolated yields. [b]For primary alcohols: reactions were conducted on 0.2 mmol scale using 2.2 equiv of SulfoxFluor, 4.0 equiv of NaN₃ and 1.8 equiv of DBU. [c]Reaction was conducted on 0.2 mmol scale using 2.5 equiv of SulfoxFluor, 5.0 equiv of NaN₃, and 1.8 equiv of DBU. [d]For secondary alcohols: reactions were conducted on 0.2 mmol scale using 2.8 equiv of SulfoxFluor, 2.0 equiv of NaN₃, and 4.0 equiv of DBU. [e]The abbreviation e.s. refers to enantiospecificity, e.s. = (e.e. of **3**)/(e.e. of **2**) × 100%. [f]Reaction was conducted on a 0.2 mmol scale using 1.05 equiv of **1j**, 4.0 equiv of NaN₃, and 1.2 equiv of DMAP in DMF (0.2 M) at rt for 12 h. [g]Epimer ratio >20:1. [h]Reaction was conducted on a 0.4 mmol scale. [i]Reaction was conducted on a 0.16 mmol scale.

(a) Gram-scale synthesis and further elaboration of alkyl azide **3ai**

(b) One-pot reaction

**Fig. 3 Synthetic applications. a** Gram-scale synthesis of alkyl azide 3ai and its further elaboration via Click reaction. **b** One-pot deoxyazidation and subsequent Click reaction. Conditions A: CuSO$_4$•5H$_2$O (1 mol%), sodium ascorbate (10 mol%), tBuOH/H$_2$O = 1:1, rt, 24 h.

(2,4-Cl$_2$C$_6$H$_3$O)$_2$P(O)Cl in the activation of normal secondary alcohols.

**Synthetic applications**. To further demonstrate the synthetic utility of the current deoxyazidation protocol, we carried out the gram-scale synthesis. As shown in Fig. 3a, the deoxyazidation reaction of dansyl chloride derivative **2ai** was carried out under standard conditions. This reaction proceeded well and afforded the desired azide **3ai** in 94% yield. Remarkably, the azide product **3ai** could be converted to triazole **9** under copper catalysis[59] in nearly quantitative yield, which significantly increased the complexity of the molecule and demonstrated the potential application of this azidation protocol in drug discovery. Furthermore, indometacin derivative **2at** was subjected to the deoxyazidation reaction and subsequent click reaction in one pot, and triazole **10** was obtained in 83% overall yield (Fig. 3b; for details, see the Supplementary Methods).

**Experimental investigation of a reaction mechanism**. Control experiments were performed to investigate the reaction mechanism (Fig. 4). Alcohol **2a** reacted under standard conditions to give azide **3a** in 84% yield (determined by $^{19}$F NMR; Fig. 4, Eq 1). However, when DBU was not added, the expected product **3a** was not detected and the alcohol **2a** remained, along with the complete consumption of SulfoxFluor (Fig. 4, Eq 2). This result indicates that SulfoxFluor itself could react with NaN$_3$. Further experiments showed the pre-formed sulfonimidoyl azide intermediate was not able to undergo the desired deoxyazidation reaction (Fig. 4, Eqs 3 and 4). Based on the above-mentioned experiments, a plausible reaction mechanism is proposed for the deoxyazidation of alcohols with SulfoxFluor (Fig. 4, Eq 5). The activated alcohol (by DBU) reacts with SulfoxFluor to form sulfonimidate ester **11**, which undergoes a nucleophilic displacement of the sulfonimidate group by azide ion to give the corresponding alkyl azide. The success of the azidation reaction lies in better nucleophilicity of azide ion over that of fluoride ion.

(a) Mechanistic experiments

(b) Proposed mechanism

**Fig. 4 Mechanistic experiments. a** The comparison of the standard experiment and the control experiments. Eq 1 refers to the deoxyazidation reaction conducted under the standard conditions. Eq 2 refers to the control experiment performed in the absence of DBU. Eq 3 refers to the control experiment performed via reverse addition of the reactants. **b** Proposed mechanism of competitive consumption of SulfoxFluor by NaN$_3$ (Eq 4) and the desired deoxyazidation of alcohols (Eq 5). ND not detected.

## Discussion

We have developed a general protocol for the direct deoxyazidation of alcohols with NaN$_3$, which provides a powerful tool to synthesize structurally diverse alkyl azides from readily available alcohols under mild conditions. Our previously developed SulfoxFluor reagent[60,61] plays an important role in this efficient deoxyazidation reaction. To our knowledge, the substrate scope and functional group tolerance of this method are superior to those of other deoxyazidation reactions (starting from alcohols) reported to date. Moreover, we have shown that this method can be applied in the late-stage modification of natural products and pharmaceutically relevant molecules, showcasing that this protocol promises to find practical applications in life sciences and related fields. Further exploration in this direction is underway in our laboratory.

## Methods

**General**. The general procedures for deoxyazidation of primary alcohols **2** with SulfoxFluor **1a** are as follows. In a typical experiment, into a 25-mL Schlenk tube (glass) were sequentially added 2-(naphthalen-1-yl)ethan-1-ol **2c** (34.4 mg, 0.2 mmol), SulfoxFluor (152.9 mg, 0.44 mmol, 2.2 equiv), NaN$_3$ (52.0 mg, 0.8 mmol), DMF (2.0 mL), and DBU (54 μL, 0.36 mmol, 1.8 equiv) under N$_2$ atmosphere. The mixture was stirred at room temperature for 12 h. Then water (2–5 mL) was added and the mixture was extracted with Et$_2$O (3 × 2 mL). The combined organic layers were dried over Na$_2$SO$_4$, filtered, concentrated under reduced pressure, and purified by chromatography on silica gel to afford alkyl azide **3c** (37.7 mg, 96%). The deoxyazidation of secondary alcohols **2** with SulfoxFluor **1a** were carried out similarly and the procedures are presented in Supplementary Methods.

## Data availability

The authors declare that the data supporting the findings of this study are available within the article and its Supplementary Information files. For the experimental procedures, and spectroscopic and physical data of compounds, see Supplementary Methods. For NMR analysis of the compounds in this article, see Supplementary Figs. 1–134. The CCDC 2005774 [https://doi.org/10.5517/ccdc.csd.cc25b5d3] contains the crystallographic data for compound **3v** (Supplementary Fig. 139;

Supplementary Table 6). These data can be obtained free of charge from the Cambridge Crystallographic Data Center (www.ccdc.cam.ac.uk).

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

## Acknowledgements

Financial support for this work by the National Key Research and Development Program of China (2021YFF0701700 and 2016YFB0101200), the National Natural Science Foundation of China (21632009), the Key Programs of the Chinese Academy of Sciences (KGZD-EW-T08), the Key Research Program of Frontier Sciences of CAS (QYZDJ-SSW-SLH049), and Shanghai Science and Technology Program (18JC1410601) is acknowledged. J.G. thanks Jie Sun (SIOC) for assistance with the X-ray crystallographic analysis.

## Author contributions

J.H. conceived the project. J.H., J.G., X.W., and C.N. designed the experiments, analyzed the data, and co-wrote the manuscript. J.G and X.W. performed synthetic and mechanistic experiments. X.(L.)W. tested the e.e. value. J.H., J.G., X.W., and C.N. discussed the results and commented on the manuscript.

## Competing interests

The authors declare no competing interests.
