## [Peer Review File · Nature Communications]

Reviewers' comments:

Reviewer #1 (Remarks to the Author):

This manuscript describes a new method for preparing alkyl azides from alcohols using sodium azide and N-tosyl-4-chlorobenzenesulfonimidoyl fluoride (SulfoxFluor) in the presence of DBU in DMF. The compounds appear to be well-characterized. In the future, the authors should include a Table of Contents and page numbers in the supporting information. Although the methodology reported this manuscript uses readily available NaN₃, it also uses SulfoxFluor which is not commercially available (unless you count Chemshuttle which apparently sells 10 g of the compound for \$1,000 USD with a 12 week wait time) or requires several steps to make. Consequently, I do not see this methodology be widely adopted. Moreover, there are already many ways to make alkyl azides and I would not say that the synthesis of alkyl azides represents a major challenge/problem in organic chemistry. In my opinion, although this methodology appears to be, in some ways, an improvement over current methodology, it is not a significant enough advancement in the field to warrant publication in Nature Communications. This manuscript would be more suited for a journal specializing in synthetic organic chemistry such as The Journal of Organic Chemistry.

Reviewer #2 (Remarks to the Author):

Hu and his group developed an operationally simple alkyl azides forming process using sulfoxfluor as an activator. Fifty structurally alcohols were applied in this article including pharmaceutically molecules in good or excellent yields. The method also has good selectivity toward multiple alcohols and undergoes a stereospecific process which can be applied in the late-stage modification of natural products and pharmaceutically molecules. The manuscript is well written and documented and I expect that this approach will find great utility in both academic and industrial settings. Authors claimed that "the currently known NaN₃-based deoxyazidation methods ... suffer from the high reaction temperatures and/or narrow substrate scope." However, the same synthetic strategy was reported (see ref 30: A simple one-pot procedure for the direct conversion of alcohols to azides via phosphate activation. *Org. Lett.*, 2, 1959-1961 (2000).), in which bis(2,4-dichlorophenyl) chlorophosphite was used as an activator, DMAP was used as a base and the reaction was also performed under room temperature. Thus, this reaction need to show some examples which have some advantages comparing to the reported method (*Org. Lett.*, 2, 1959-1961 (2000)).

Additional some suggestions and corrections:

1. Are other inorganic bases suitable in the reaction?

2. Can some other functional group be tolerated in this reaction, such as sulfur alcohol or phenolic hydroxyl?

3. 1a should be 2a and “sufoxfluor” should be “sulfoxfluor” in Table 1.

4. Attention should be paid to typesetting in Table 1.

5. The structure of 3b is wrong in Table 2.

Overall, I recommend that this manuscript could be accepted for publication in Nature Communications pending the inclusion of these major modifications.

Reviewer #3 (Remarks to the Author):

This work by Hu and co-workers describes the general and practical method for the deoxylazidation of alcohols. This method is enabled by SulfoxFluor which had been previously developed by Hu's lab. There are several important features of this protocol, 1) mild conditions; 2) broad functional group compatibility; 3) late-stage modification of important bioactive molecules; 4) high stereospecificity with chiral alcohols. In addition, the quality of the SI is very high with clean spectra. Although there are many known deoxylazidation methods, this method is still superior and will become the most classic one which will find broad applications in organic synthesis. For these reasons, this reviewer suggests the acceptance of this paper by Nature Communications after minor revisions:

1) Table 2. Structure of 3b is wrong.

2) Just for the curiosity, is this reaction compatible with primary and secondary amines? What will be generated in presence of these two types of amines?

II. Response to Reviewers

Reviewer #1

Comment 1: This manuscript describes a new method for preparing alkyl azides from alcohols using sodium azide and *N*-tosyl-4-chlorobenzenesulfonimidoyl fluoride (SulfoxFluor) in the presence of DBU in DMF. The compounds appear to be well-characterized.

Our Response: We thank the reviewer for this summary of our work.

Comment 2: In the future, the authors should include a Table of Contents and page numbers in the supporting information.

Our Response: Following this suggestion, we have included a Table of Contents and page numbers in the revised supporting information.

Comment 3: Although the methodology reported this manuscript uses readily available NaN_3 , it also uses SulfoxFluor which is not commercially available (unless you count Chemshuttle which apparently sells 10 g of the compound for \$1,000 USD with a 12 week wait time) or requires several steps to make. Consequently, I do not see this methodology be widely adopted.

Our Response: Very recently, we have developed the modified and scalable synthesis of SulfoxFluor (*Org. Process Res. Dev.* 2022, ASAP, <https://pubs.acs.org/doi/10.1021/acs.oprd.1c00431>). By using this modified method, SulfoxFluor could be prepared from the easily available 4-chlorobenzenesulfonyl chloride and chloramine-T trihydrate in high overall yield with simple purification techniques.

At present, we can prepare 10 kilograms in one batch. The total cost is about \$0.15 USD per gram of SulfoxFluor. We have provided this reagent to our customers

at affordable prices (\$3-6 USD/gram, including handling fees) (two copies of the purchase orders are provided as the Supplementary Files for Review, purchase order 1 and purchase order 2; \$1 USD = ¥6.37 CNY).

Currently, SulfoxFluor (CAS No.: 2143892-50-4) is also commercially available from Daicel Chiral Technologies (China) Co., Ltd. (<https://www.daicelchiraltech.cn/en/reagents/detail.aspx?id=6302>). The price for the package of 1 gram is about \$47 USD, and the price for large package will be much lower.

SulfoxFluor is an important deoxyfluorination reagent, featured by its crystalline state, easy availability, and high fluorination/elimination selectivity. This reagent has been used by Wang *et al.* for the deoxyfluorination of a challenging alcohol (Ref: Tang, H.; Cheng, J.; Liang, Y.; Wang, Y. *Eur. J. Med. Chem.* **2020**, *197*, e112323). In a recent review, Stéphane Caron from Pfizer has listed SulfoxFluor as one of the leading nucleophilic fluorinating reagents (Ref. Table 2 in: Caron, S. *Org. Process Res. Dev.* **2020**, *24*, 470–480). And very recently, SulfoxFluor has been included in *e-EROS* by the editorial board André Charette, Jeffrey Bode, Tomislav Rovis, and Ryan Shenvi (Ref. Guo, J.; Hu, J. *N-Tosyl-4-chlorobenzenesulfonimidoyl Fluoride* (SulfoxFluor) in: *Encyclopedia of Reagents for Organic Synthesis*. DOI: 10.1002/047084289X.rm02363).

Therefore, we believe that this deoxyazidation methodology will be widely adopted due to the easy availability and privileged application of SulfoxFluor. We have changed Ref. 61 and added Ref. 62 in the Revised Manuscript ;

61. Liu, R. *et al.* Modified and Scalable Synthesis of *N*-tosyl-4-chlorobenzenesulfonimidoyl fluoride (SulfoxFluor): direct Imidation of sulfinyl chlorides with chloramine-T trihydrate. *Org. Process Res. Dev.* **26**, <https://pubs.acs.org/doi/10.1021/acs.oprd.1c00431> (2022).

62. Zhou, X. *et al.* Method for preparing a deoxyfluorination reagent. Chinese Patent Application. CN 113717087 A (2020).

Comment 4: In my opinion, although this methodology appears to be, in some ways, an improvement over current methodology, it is not a significant enough advancement in the field to warrant publication in Nature Communications.

Our Response: Following this suggestion, we have compared our methodology with the state-of-the-art method using bis(2,4-dichlorophenyl) chlorophosphate ((2,4-Cl₂C₆H₃O)₂P(O)Cl)/NaN₃/DMAP (Ref. *Org. Lett.* **2000**, 2, 1959). Gratifyingly, we have found that our methodology is amenable with both electron-deficient primary alcohols such as CF₃CH₂OH (**2c**) and normal secondary alcohols such as **2r**. However, the (2,4-Cl₂C₆H₃O)₂P(O)Cl system is less efficient than our SulfoxFluor system in the conversion of these two kinds of alcohols.

Results on the deoxygenation of alcohols **2c** and **2r** have been added into the Revised Manuscript (both in the main text and in Tables 3 and 4).

For details, see our Response to Comment 2 from Reviewer 2.

Reviewer #2

Comment 1: Hu and his group developed an operationally simple alkyl azides forming process using SulfoxFluor as an activator. Fifty structurally alcohols were applied in this article including pharmaceutically molecules in good or excellent yields. The method also has good selectivity toward multiple alcohols and undergoes a stereospecific process which can be applied in the late-stage modification of natural products and pharmaceutically molecules. The manuscript is well written and documented and I expect that this approach will find great utility in both academic and industrial settings.

Overall, I recommend that this manuscript could be accepted for publication in Nature Communications pending the inclusion of these major modifications.

Our Response: We thank Reviewer 2 for highlighting the novelty of our study and for the positive recommendations.

Comment 2: Authors claimed that “the currently known NaN₃-based deoxygenation

methods ... suffer from the high reaction temperatures and/or narrow substrate scope.” However, the same synthetic strategy was reported (see ref 30: A simple one-pot procedure for the direct conversion of alcohols to azides via phosphate activation. *Org. Lett.*, **2**, 1959-1961 (2000).), in which bis(2,4-dichlorophenyl) chlorophosphate was used as an activator, DMAP was used as a base and the reaction was also performed under room temperature. Thus, this reaction need to show some examples which have some advantages comparing to the reported method (*Org. Lett.*, **2**, 1959-1961 (2000)).

Our Response: Following this suggestion, we have compared our methodology with the state-of-the-art method using bis(2,4-dichlorophenyl) chlorophosphate ((2,4-Cl₂C₆H₃O)₂P(O)Cl)/NaN₃/DMAP (Ref. Yu, C.; Liu, B.; Hu, L. *Org. Lett.* **2000**, **2**, 1959). Gratifyingly, we have found that our methodology is amenable with both electron-deficient primary alcohols such as CF₃CH₂OH (**2c**) (Scheme R1a) and normal secondary alcohols such as **2r** (Scheme R1b). However, the (2,4-Cl₂C₆H₃O)₂P(O)Cl system is less efficient than our SufoxFuor system in the conversion of these two kinds of alcohols.

Scheme R1. Deoxyzidation with the reported (2,4-Cl₂C₆H₃O)₂P(O)Cl system

Results on the deoxyzidation of alcohols **2c** and **2r** have been added into the

Revised Manuscript (both in the main text and in Tables 3 and 4). In the main text, we have added the following results:

It is interesting to note that the use of bis(2,4-dichlorophenyl) chlorophosphate ((2,4-Cl₂C₆H₃O)₂P(O)Cl, **1j**)/NaN₃/DMAP, a state-of-the-art method for deoxyazidation of alcohols at room temperature,³⁰ failed to convert 2,2,2-trifluoroethanol (**2c**) into azide **3c** (Table 3, entry 10; for details, see the Supplementary Methods).

Our synthetic method is also advantageous over the previously reported deoxyazidation system (2,4-Cl₂C₆H₃O)₂P(O)Cl/NaN₃/DMAP in converting normal secondary alcohols to organoazides. For example, starting from alcohol **2r**, the use of SulfoxFluor could afford the corresponding azide **3r** in nearly quantitative yield, whereas the use of (2,4-Cl₂C₆H₃O)₂P(O)Cl under reported standard conditions³⁰ provided **3r** in low yield (21%) even prolonging the reaction time to 12 h. In the latter case, both the bis(2,4-dichlorophenyl) phosphate intermediate and unreacted alcohol **2r** were isolated (for details, see the Supplementary Methods), indicating the low efficiency of (2,4-Cl₂C₆H₃O)₂P(O)Cl in the activation of normal secondary alcohols.

Comment 3: Are other inorganic bases suitable in the reaction?.

Our Response: Inorganic bases K₂CO₃ and K₃PO₄ are not suitable in this deoxyazidation reaction probably due to their poor solubility. Nevertheless, organic bases pyridine and Et₃N are also not suitable probably due to their relative weak basicity. DBU is the most suitable base in terms of its reactivity and price.

Based on our mechanistic investigations, SulfoxFluor is also reactive towards NaN₃. In this context, a strong organic base that can promote the fast formation of the sulfonimidate ester intermediate (such as DBU) is beneficial for the deoxyazidation of alcohols.

Comment 4: Can some other functional group be tolerated in this reaction, such as sulfur alcohol or phenolic hydroxyl?

Our Response: In a separate research (unpublished results), we have found that under

similar conditions, the anions of thiols, phenols, carboxylic acids and soft carbon acids could undergo substitution reactions with alcohols. Therefore, we predict that functional groups such as sulfur alcohol or phenolic hydroxyl should compete with NaN_3 in the conversion of alcohols.

Comment 5: **1a** should be **2a** and “sufoxfleur” should be “sulfoxfluor” in Table 1.

Our Response: We have corrected **1a** to **2a** and “SufoxFleur” to “SulfoxFluor” in Table 1. We have also corrected “SufoxFleur” to “SulfoxFluor” in Table 2.

Comment 6: Attention should be paid to typesetting in Table 1.

Our Response: We have carefully revised Table 1, and set the equation in the middle of the Table, changed “Pyridine” to “pyridine”, and removed “Ar = 4-chlorophenyl” in the footnote.

Comment 7: The structure of **3b** is wrong in Table 2.

Our Response: We have corrected the structure of **3b** in Table 2.

We have also removed “Ar = 4-chlorophenyl” in the footnote.

Reviewer #3

Comment 1: This work by Hu and co-workers describes the general and practical method for the deoxylazidation of alcohols. This method is enabled by SulfoxFluor which had been previously developed by Hu’s lab. There are several important features of this protocol, 1) mild conditions; 2) broad functional group compatibility; 3) late-stage modification of important bioactive molecules; 4) high stereospecificity with chiral alcohols. In addition, the quality of the SI is very high with clean spectra. Although there are many known deoxylazidation methods, this method is still superior and will become the most classic one which will find broad applications in organic synthesis. For these reasons, this reviewer suggests the acceptance of this paper by Nature Communications after minor revisions.

Our Response: We thank Reviewer 3 for highlighting the novelty of our study and for the positive recommendations.

Comment 2: Table 2. Structure of **3b** is wrong.

Our Response: We have corrected the structure of **3b** in Table 2.

Comment 3: Just for the curiosity, is this reaction compatible with primary and secondary amines? What will be generated in presence of these two types of amines?

Our Response: This reaction is not compatible with primary and secondary amines. As previously reported, sulfonimidoyl fluorides could undergo SuFEx reaction with primary and secondary amines to form sulfonimidamides (Refs: Richards-Taylor, C. S. et al. *J. Org. Chem.* **2017**, 82, 9898; Gao, B. et al. *Angew. Chem. Int. Ed.* **2018**, 57, 1939; Greed, S. et al. *Chem. Eur. J.* **2020**, 26, 12533).

III. Additional Changes

- 1) In the author list, we have added Xiu Wang as a coauthor as she put much effort in carrying out additional experiments and doing the revision work (especially in the preparation of $(2,4\text{-Cl}_2\text{C}_6\text{H}_3\text{O})_2\text{P}(\text{O})\text{Cl}$, which turned out to be challenging and very time-consuming). The first author Junkai Guo has left my research group; and based on the effort of Xiu Wang, Junkai Guo confirmed that Xiu Wang should be added as a co-author with equal contribution as him.
- 2) In the main text, in several places, we have corrected “sulfoximidoyl” to “sulfonimidoyl”.
- 3) In the main text, we have changed “supporting information” to “Supplementary Methods”.
- 4) In the section of Data availability, we have changed “Supplementary Figs 1–128” to “Supplementary Figs 1–134”, and “Supplementary Fig 133” to “Supplementary Fig 139”.
- 5) In the section of Acknowledgements, we have added a new grant number

“2021YFF0701700”.

- 6) In the section of Author contributions, we have added a description on the contribution of Xiu Wang.
- 7) In the Supplementary Information, we have added the methods for the deoxyzidation of alcohols **2c** and **2r** with bis(2,4-dichlorophenyl) chlorophosphate (Subsection 6 of the Supplementary Methods) and the NMR spectra of the corresponding compounds (Supplementary Figs 129-134).

REVIEWERS' COMMENTS

Reviewer #1 (Remarks to the Author):

Making alkyl azides is not a major challenge in organic chemistry and it is NOT a significant problem in organic synthesis. So although this methodology appears to be, in some ways, an improvement over current methodology, this manuscript should not be published in Nat. Comm. It should be sent to JOC.

Reviewer #2 (Remarks to the Author):

This revised manuscript is significantly improved from the version that I reviewed previously. The authors have made the necessary changes to the original manuscript and, in my opinion, the revised manuscript is suitable for publication in Nature Communications.

Reviewer #3 (Remarks to the Author):

All of my previous comments/questions have been properly addressed.

I am also pleased to see that the revised manuscript cites the authors' recent progress on the improved and scalable synthesis of SulfoxFluor reagent (Ref. 61), which further enables this SulfoxFluor-mediated deoxyfluorination method to find wide applications in organic synthesis. Indeed, SulfoxFluor is now commercially available with reasonable cost. Overall, I recommend the publication of this nice chemistry in Nature Communications.